# Association between Environmental Cadmium Exposure and Osteoporosis Risk in Postmenopausal Women: A Systematic Review and Meta-Analysis

**DOI:** 10.3390/ijerph20010485

**Published:** 2022-12-28

**Authors:** Carlos Tadashi Kunioka, Maria Conceição Manso, Márcia Carvalho

**Affiliations:** 1FP-I3ID, FP-BHS, University Fernando Pessoa, 4249-004 Porto, Portugal; 2Western Paraná State University (UNIOESTE), Cascavel 85819-110, Paraná, Brazil; 3Faculty of Health Sciences, University Fernando Pessoa, 4200-150 Porto, Portugal; 4LAQV, REQUIMTE, University of Porto, 4050-313 Porto, Portugal; 5Associate Laboratory i4HB-Institute for Health and Bioeconomy, Faculty of Pharmacy, University of Porto, 4050-313 Porto, Portugal; 6UCIBIO-REQUIMTE, Laboratory of Toxicology, Department of Biological Sciences, Faculty of Pharmacy, University of Porto, 4050-313 Porto, Portugal

**Keywords:** environmental cadmium, bone density, osteoporosis, risk factor, menopause, women, meta-analysis

## Abstract

Osteoporosis is a common and serious health issue among postmenopausal women. We conducted a systematic review and meta-analysis study to determine whether environmental exposure to cadmium (Cd) is a risk factor for postmenopausal osteoporosis. A PROSPERO-registered review of the literature was performed on studies evaluating the relationship between urinary Cd (UCd) concentration, an indicator of long-term Cd exposure, and bone mineral density or osteoporosis in women aged 50 years and older. PubMed, Embase, Science Direct, Web of Science, and B-on databases were searched for articles published between 2008 and 2021. The association between UCd levels and osteoporosis risk was assessed by pooled odds ratio (OR) and 95% confidence interval (CI) using random-effect models. Ten cross-sectional studies were included in the qualitative analysis, of which five were used for meta-analysis. We separately assessed the risk of osteoporosis in women exposed to Cd at low environmental levels (*n* = 5895; UCd ≥ 0.5 μg/g creatinine versus UCd < 0.5 μg/g creatinine) and high environmental levels (*n* = 1864; UCd ≥ 5 μg/g creatinine versus UCd < 5 μg/g creatinine). The pooled OR for postmenopausal osteoporosis was 1.95 (95% CI: 1.39–2.73, *p* < 0.001) in the low exposure level group and 1.99 (95% CI: 1.04–3.82, *p* = 0.040) in the high exposure level group. This study indicates that environmental Cd exposure, even at low levels, may be a risk factor for osteoporosis in postmenopausal women. Further research based on prospective studies is needed to validate these findings.

## 1. Introduction

Osteoporosis is a global disease that has an important impact on the health and economy of countries [1,2]. It is estimated that more than 500 million people worldwide suffer from osteoporosis [3], and the disease’s incidence is rising as the population ages and lifestyle habits change [4,5]. This bone disorder is characterized by a decreased bone mineral density (BMD) and deterioration of skeletal microarchitecture, predisposing to low-impact, fragility fractures [6]. Osteoporotic fractures have a significant negative impact on patient′s quality of life, as well as increased morbidity, mortality, and disability [1,7], resulting in enormous healthcare costs [8,9]. Certain risk factors associated with the development of osteoporosis and associated fractures have been identified, including non-modifiable ones such as advanced age, female gender, menopause, genetics, and modifiable ones such as excessive alcohol intake, smoking, low physical activity, underweight, and chronic use of glucocorticoids [10,11,12]. Human exposure to heavy metals, such as cadmium (Cd), has also been shown to affect bone metabolism, predisposing to an increased prevalence of osteoporosis [13,14]. As the global burden of osteoporosis grows, it is critical to identify risk factors linked to the loss of bone mass, particularly those that can be avoided, such as Cd exposure, thus helping to control the incidence of this complex condition.

Cd is a naturally occurring heavy metal, but human exposure to it is increased by anthropogenic activity [15]. One of the primary sources of Cd pollution is industrial activity. Cd can be found in landfills from smelters, iron and steel plants, electroplating, and battery manufacturing industries, whereas mining is a common source of Cd in water [16]. Additionally, the use of Cd-containing fertilizers is of particular concern due to metal uptake from soil and bioaccumulation by crops [17]. Excessive amounts of Cd have been shown to enter the food chain system and the general population is primarily exposed to Cd through contaminated food (mostly cereals and leafy vegetables) and water [18]. Additionally, Cd can enter the human body via other sources, including tobacco smoking (because tobacco plants absorb Cd from the soil and concentrate it in the leaves) and occupational exposure to Cd fumes or dust in the workplace [19].

Several adverse health effects of Cd exposure have been reported including renal dysfunction, lung and cardiovascular diseases, bone disorders, and cancers [19,20,21]. Kidneys and bones are the primary target organs of Cd toxicity in chronic, low-level Cd exposure patterns that are more common in the general population. Cd accumulates in the kidneys, mainly in the proximal tubular epithelial cells, and high levels cause renal injury [19,22]. Because of its low clearance rate, Cd has an exceptionally long half-life of 10–30 years. The toxic effect of high-level Cd exposure on bone became evident with the outbreak of Itai-Itai disease in Japan in the 1950s [19]. Several subsequent population-based studies have shown that Cd exposure, even at low environmental levels, may result in decreased bone mass and an increased risk of osteoporosis and bone fractures [14,23,24], though some studies reported no or a weak association between low-level Cd exposure and bone disorders [25,26,27].

To date, previous meta-analysis studies have only investigated the relationship between Cd exposure and bone disorders (osteopenia, osteoporosis [28,29], and any fracture [30]) in the general population. However, women are more likely than men to suffer from osteoporosis and associated fractures [2], which worsens with the onset of menopause [12]. As a result, postmenopausal women are a more relevant population to study this association. Herein, we performed a systematic review and meta-analysis study to (1) summarize the evidence on the relationship between environmental exposure to Cd and BMD or osteoporosis in the postmenopausal women population, and (2) determine whether environmental Cd exposure is a risk factor for postmenopausal osteoporosis.

## 2. Methods

### 2.1. Literature Search

This literature review was registered in PROSPERO (CRD42021241377) and conducted in accordance with the Preferred Reporting Items for Systematic Reviews and Meta-Analyses (PRISMA) guidelines [31]. Databases of PubMed, Embase, Science Direct, Web of Science and B-on were systematically searched for relevant studies published between January 2008 and December 2021 using the search terms: (cadmium AND women AND bone density AND environmental exposure). Grey literature was searched using Google Scholar and the OpenGrey online database. To find additional relevant studies, a manual search of all references of the included studies was also performed. A detailed PubMed search strategy is provided in Appendix A.

### 2.2. Study Selection and Data Extraction

Duplicates were found and removed from the initial search results. Two authors (C.T.K. and M.C.) independently reviewed the titles and abstracts of all studies to screen for eligibility. The inclusion criteria were as follows: (1) observational studies; (2) studies including women aged 50 years and older; (3) studies that provided information on the relationship between UCd levels and BMD and/or osteoporosis risk; (4) papers published in the English language. The exclusion criteria were as follows: (1) review articles, brief communications, or case reports; (2) studies based on data from populations occupationally exposed to Cd; (3) articles that only reported findings on men or did not allow to separate results in men and women; and (4) in vitro or laboratory studies. Any disagreement about the inclusion and exclusion of studies was resolved by a third author (M.C.M.). For the meta-analysis, retrieved eligible studies that reported UCd levels and prevalence of osteoporosis in women aged ≥ 50 years were included. Urinary levels of Cd were used as a measure of cumulative long-term exposure to this metal [32] and osteoporosis was defined based on the WHO criteria of a T score of −2.5 or lower [33].

All authors independently reviewed the full text of the articles for the final study collection based on the inclusion criteria. If more than one paper based on the same dataset was published, the most recent paper or the one with the best outcome assessment was included. When required, the authors of the original studies were contacted to obtain additional data.

The data extracted from each study included first author’s last name, publication year, country and design of the study, sample size, age, smoking status, UCd measurement method, urine sample type, UCd levels, BMD measurement technique and body region examined, BMD levels, osteoporosis prevalence, adjusted variables, and relevant study findings.

### 2.3. Assessment of Methodological Quality

Two reviewers (C.T.K. and M.C.) independently assessed the quality of the included studies using the Joanna Briggs Institute (JBI) critical appraisal checklist for cross-sectional studies [34], which assesses study quality based on eight criteria, including bias, confounding, the validity of exposure and outcome measurement, and the validity of methods of analyses. Each item was labeled as “yes”, “no” or “unclear”. A third reviewer resolved any disagreements (M.C.M.). The final score of each article was calculated based on the percentage of positive answers (‘yes’), with scores greater than 70% indicating a ‘low’ risk of bias, scores ranging from 50% to 69% indicating a ‘moderate’ risk of bias, and scores less than 49% indicating a ‘high’ risk of bias.

### 2.4. Statistical Analysis

A random-effects model was used for the meta-analysis. Odds ratio (OR) with a 95% confidence interval (CI) was estimated for each study and used to assess the strength of the association between UCd levels and osteoporosis risk. The significance of the pooled OR was determined by the Chi-square test. Heterogeneity between the studies was assessed by a Chi-square-based Q-test and I^2^ statistics and was considered significant if the *p*-value for the Q-test was <0.10 or I^2^ > 50% [35]. Heterogeneity can be quantified as “low”, “moderate”, and “high”, with upper limits of 25%, 50% and 75% for I^2^, respectively. The Begg’s funnel plot and the Egger’s test were used to evaluate publication bias. Forest plot was used to show the effect measures of each included study and the pooled effect measures. All statistical analyses were performed using IBM^®^ SPSS© Statistics vs.28.0 (IBM Corp. released 2021, Armonk, NY, USA). Unless indicated otherwise, all statistical tests were considered significant at *p* < 0.05.

## 3. Results

### 3.1. Study Selection

Figure 1 depicts the PRISMA flowchart used for the selection process. The initial database search retrieved 192 articles, with 122 remaining after duplicates were removed. Studies were then eliminated based on their title and/or abstract. The full texts of the remaining 58 publications were carefully reviewed and selected according to the inclusion and exclusion criteria. Finally, ten articles that fulfilled the eligibility criteria were included in the present work for qualitative analysis and five for meta-analysis (Figure 1).

### 3.2. Study Characteristics

General characteristics of the included studies are described in Table 1 and are summarized below. Additional data inspection and conversion were performed to unify the database. Due to data overlap in serial articles by Chen et al. [36,37,38,39,40], only the one most relevant to this study was included [39]. The authors of three original studies [18,23,39] kindly provided additional unpublished data.

Ten studies published from 2008 to 2021 and a total of 9205 women were included in the systematic review. Three studies were conducted in Sweden [41,42,43], two in China [23,39], one in Japan [25], one in South Korea [26], one in the USA [44], one in Australia [14], and one in Thailand [18]. All studies had a cross-sectional design. The women’s ages ranged from 50 to 90 years and non-smokers outnumbered smokers in all studies. Cd concentrations in urine samples were determined using atomic absorption spectrometry (AAS) in five studies [18,25,26,39,44] or inductively coupled plasma mass spectrometry (ICP-MS) in the other five [14,23,41,42,43]. UCd values were adjusted to the urine creatinine concentrations and ranged from less than 0.065 up to 16.17 μg/g creatinine. Dual-energy X-ray absorptiometry (DXA) is the gold standard for measuring BMD [12] and was used in all studies except one, which used ultrasound bone densitometry (USBD) [26]. The skeletal sites for BMD measurements differed between studies and included the forearm in three [23,25,39]; the wrist in two [42,43]; the calcaneus in two [18,26]; the hip in three [14,41,44]; and the proximal femur and lumbar spine in two [14,41].

Six studies provided data on the relationship between UCd levels and osteoporosis prevalence [18,23,26,39,41,44], but only five with a total of 7759 women provided UCd stratified data that allowed for comparisons between studies and thus were included in the meta-analysis. There were clear differences in UCd concentrations between studies, with North America [44] and Europe [41] presenting lower UCd concentrations than Asian countries, including China [23,39], South Korea [26], and Thailand [18]. As a result, these studies were divided into two groups for further statistical analysis based on reported UCd levels: the low-level Cd exposure group (with a cut-off value of 0.5 μg/g creatinine) comprising the studies conducted in the USA and Europe with a total of 5895 women; and the high-level Cd exposure group (with a cut-off-value of 5 μg/g creatinine) comprising three studies conducted in Asian countries with a total of 1864 women. Because everyone in the population is exposed to Cd through food, a reference group with no exposure cannot be defined; thus, two exposure categories were used for the low-level (UCd ≥ 0.5 μg/g creatinine versus UCd < 0.5 μg/g creatinine) and high-level (UCd ≥ 5 μg/g creatinine versus UCd < 5 μg/g creatinine) groups.

**Table 1 ijerph-20-00485-t001:** Characteristics of studies included in the systematic review and meta-analysis evaluating the effects of environmental cadmium exposure on bone mineral density or osteoporosis in women aged 50 and older from 2008 to 2021.

Study	Country/Study Design	Sample Size	Age (Years)/Menopausal Status	Smoking Status	UCd Measurement Method/Sample Type	UCd (µg/g Creatinine)	BMD Measurement Technique/Location	BMD (g/cm^2^)	Prevalence of Osteoporosis (%)	Adjusted Variables	Relevant Findings
Gallagher et al., 2008 ^a^ [44]	USA/Cross-sectional	*n* = 3207 (UCd < 0.5, *n* = 870; UCd 0.5–1.0, *n* = 1201; UCd > 1.0, *n* = 1136)	67 (50–90)/ Not specified	Never: 61%, Ever: 39%	AAS/ Spot urine	0.96 (0.007–16.17)	DXA/ Hip	Not specified	UCd < 0.5: 14.6%; UCd 0.5–1.0: 21.4%; UCd > 1.0: 24.0%	Age, race, income, underweight, and smoking status	UCd was significantly associated with a greater risk for osteoporosis at levels ≤1.0 μg/g creatinine.
Engstrom et al., 2009 [42]	Sweden/Cross-sectional	*n* = 85 (Low UCd, *n* = 45); High UCd, *n* = 40)	58 (54–63)/Postmenopausal: 100%	Low UCd: Never: 76%, Ever: 24% High UCd: Never: 30%, Ever: 70%	ICP-MS/ Not specified	Low UCd: 0.36 (0.18–0.73); High UCd: 1.1 (0.69–1.7)	DXA/ Wrist	Low UCd: 0.45 (0.35–0.53); High UCd: 0.43 (0.31–0.54)	Not determined	No	BMD was significantly lower in the high-level Cd exposure group.
Horiguchi et al., 2010 [25]	Japan/Cross-sectional	*n* = 252 (Control area, *n* = 123; Polluted area, *n* = 129)	Control area, 54.8 ± 7.9 Polluted area, 56.6 ± 8.1/ Perimenopausal: 30.5%; Postmenopausal 49.6%	Never: 100%	Flameless AAS/ Spot urine	Control area: 3.36 ± 1.86; Polluted area: 6.30 ± 1.98	DXA/ Forearm	Control area: 0.431 ± 0.078; Polluted area: 0.423 ± 0.090	Not determined	No	BMD was not statistically different between control and Cd-polluted areas.
Suwazono et al., 2010 [43]	Sweden/Cross-sectional	*n* = 794	58 (54–63)^/^ Postmenopausal: 46%	Never: 55%, Former: 23%, Current: 22%	ICP-MS/Morning urine	0.67 (0.31–1.57)	DXA/ Wrist	0.44 (0.33–0.54)	7.7%	Age, weight, menopausal status or HRT use, and sampling season	UCd was significantly and inversely associated with T-score.
Chen et al., 2011 ^#^ [39]	China/Cross-sectional	*n* = 238 (Control area, *n* = 61; Moderate polluted area, *n* = 80; Heavy polluted area, *n* = 97)	Control area: 50–86, Moderate polluted area: 50–83, Heavy polluted area: 50–82/ Not specified	Non-smoker: 100%	GF-AAS/ Not specified	Control area: 3.1 ± 2.3; Moderate polluted area: 5.2 ± 3.5; Heavy polluted area: 11.7 ± 7.7	DXA/ Forearm	Control area: 0.68 ± 0.01; Moderate polluted area: 0.64 ± 0.008; Heavy polluted area: 0.60 ± 0.007	Control area: 42.6%; Moderate polluted area: 31.2%; Heavy polluted area: 64.9%	Age	BMD was significantly lower in the moderate and high Cd-exposure groups.
Engstrom et al., 2011 ^a^ [41]	Sweden/Cross-sectional	*n* = 2688 (UCd < 0.50, *n* = 2067; UCd 0.50–0.75, *n* = 449; UCd ≥ 0.75, *n* = 172)	56–69, UCd < 0.50, 63 (60–69); UCd 0.50–0.75, 64 (60–69); UCd ≥ 0.75, 63 (60–69)/Postmenopausal: 100%	UCd < 0.50, Non-smoker: 47%; UCd 0.50–0.75, Non-smoker: 74%; UCd ≥ 0.75, Non-smoker: 81%	ICP-MS/ Morning urine	UCd < 0.50: 0.30 (0.14–0.47); UCd 0.50–0.75: 0.59 (0.51–0.72); UCd ≥ 0.75: 0.87 (0.76–1.5)	DXA/ Femoral neck; Total hip and Lumbar spine (data not shown)	UCd < 0.5 µg/g: 0.89 (0.73–1.1) UCd 0.50–0.75 µg/g: 0.88 (0.69–1.1) UCd ≥ 0.75 µg/g: 0.85 (0.67–1.1)	UCd < 0.50 µg/g: 6.5%; UCd 0.5–0.75 µg/g: 13%; UCd ≥ 0.75 µg/g: 17%	Age, education, height, total fat mass, lean body mass, parity, HRT, corticosteroids use, physical activity, smoking status, alcohol intake, inflammatory joint diseases, kidney diseases, liver diseases, and malabsorption	UCd was inversely associated with BMD at the femoral neck and total hip. There was a statistically significant dose-dependent increase in osteoporosis risk across UCd groups. These associations were independent of tobacco smoking.
Kim et al., 2014 ^a^ [26]	Korea/Cross-sectional	*n* = 630 (UCd < 5, *n* = 501; UCd > 5, *n* = 129)	65.2 ± 10.9/ Postmenopausal: 88%	Non-smoker: 94%, Smoker: 6%	Flameless AAS/ Morning urine	2.9 ± 1.9	Ultrasound bone densitometer/ Calcaneus	Not specified	UCd < 5: 45%; UCd > 5: 54%	Age, smoking status, alcohol intake, BMI, diabetes, hypertension, and menopause	A high Cd body burden did not significantly increase the risk of osteoporosis.
Callan et al., 2015 [14]	Australia/Cross-sectional	*n* = 77	59.6 ± 7.0 (50–83)/Amenorrhea: 97%, Postmenopausal: 86%	Never: 65%, Former: 35%, Current: 0%, Smokers at household within last 6 months: 15%	ICP-MS/ Morning urine	0.26 (<0.065–1.03)	DXA/ Total hip; Femoral neck; Lumbar spine; Whole body	Total hip: 0.89 ± 0.14; Femoral neck: 0.76 ± 0.13; Lumbar spine: 0.99 ± 0.15; Whole body: 0.91 ± 0.10	Not determined	Age, years since last menstrual cycle, and BMI	An inverse relationship between UCd and BMD was discovered in all body regions studied. These associations were independent of tobacco smoking.
Lv et al., 2017 ^a,#^ [23]	China/Cross-sectional	*n* = 444 (Control area, *n* = 118; Polluted area, *n* = 326)	Control area: 60.5 ± 6.1; Polluted area: 59.2 ± 6.8/Postmenopausal: 100%	Non-smoker: 98.7%, Smoker: 1.3%	ICP-MS/ Morning urine	Control area: 1.92 (1.37–2.73); Polluted area: 7.20 (3.79–14.78)	DXA/ Forearm	Control area: 0.355 ± 0.078; Polluted area: 0.339 ± 0.083	UCd 0–2: 17.5%; UCd 2–5: 35.0%; UCd 5–10: 42.9%; UCd 10–20: 51.9%; UCd 20–40: 55.9%; UCd > 40: 75.0%	Age, BMI, serum albumin, smoking status, and urinary levels of calcium, NAG, α1-microglobulin, β2-microglobulin, and albumin	The prevalence of osteoporosis increased as UCd concentrations increased.
La-Up et al. 2021 ^a,#^ [18]	Thailand/Cross-sectional	*n* = 790 (UCd < 2, *n* = 230; UCd 2–4.9, *n* = 338; UCd 5–9.9, *n* = 184; UCd ≥ 10, *n* = 38)	59.9 ± 7.1/ Postmenopausal: 91%	Never: 75.8%, Former: 10.6%, Current: 9.6%	AAS/ Morning urine	3.98 ± 3.15	DXA/ Calcaneus	UCd < 2: 0.40 ± 0.07; UCd 2–4.9: 0.37 ± 0.07; UCd 5–9.9: 0.36 ± 0.08; UCd ≥ 10: 0.35 ± 0.09	UCd < 2: 13.0%; UCd 2–4.9: 26.9%; UCd 5–9.9: 34.8%; UCd ≥ 10: 47.4%	Age, BMI, and smoking status	There was a negative relationship between UCd and BMD and a positive association between UCd ≥ 10 μg/g creatinine and the prevalence of osteoporosis.

Notes: Data is presented as mean ± standard deviation or median (5th–95th percentile); ^a^ study included in meta-analysis; ^#^ data provided by authors considering only women aged 50 and older; Abbreviations: AAS, atomic absorption spectrometry; BMD, bone mineral density; BMI, body mass index; DXA, dual-energy X-ray absorptiometry; GF-AAS, graphite-furnace atomic absorption spectrometry; HRT, hormone replacement therapy; ICP-MS, inductively coupled plasma mass spectrometry; NAG, N-acetyl-β-D-glucosaminidase; OR, odds ratio; UCd, urinary cadmium.

### 3.3. Study Quality Assessment

The quality assessment of the ten included articles was assessed through the JBI critical appraisal tool and showed that all studies were of high quality (Appendix A).

### 3.4. The Association between UCd Levels and BMD

Six of the ten studies included in the systematic review found an association between UCd concentrations and BMD. Among these, UCd levels were significantly inversely correlated with BMD in 5 studies [14,18,39,41,42], while no association between UCd and BMD was found in one [25] (Table 1).

### 3.5. The Association between UCd Level and Osteoporosis

Four of the five studies included in the meta-analysis found a statistically significant positive association between UCd levels and osteoporosis outcome [18,23,41,44], while one failed to find an association [26]. However, it must be noted that in our meta-analysis the latter study was borderline significant (*p* = 0.05). Because of the above-mentioned heterogeneity in UCd concentrations across studies, the meta-analysis separately assessed the risk of osteoporosis in low- and high-level exposure groups, and the corresponding forest plots are shown in Figure 2A,B, respectively. The OR for osteoporosis in postmenopausal women whose UCd level was higher than 0.5 μg/g creatinine was 1.95 (95% CI: 1.39–2.73, *p* < 0.001) when compared to those with a UCd level less than 0.5 μg/g creatinine. The OR for osteoporosis in postmenopausal women with a UCd level higher than 5 μg/g creatinine was 1.99 (95% CI: 1.04–3.82, *p* = 0.040) when compared to those with a urinary Cd level less than 5 μg/g creatinine. The heterogeneity between studies was high in the high-level exposure group (I^2^ = 72%, *p* = 0.06) but moderate in the low-level exposure group (I^2^ = 45%, *p* = 0.16).

### 3.6. Publication Bias

There was no evidence of publication bias for the studies included in meta-analysis as illustrated by the symmetrical distribution of funnel plot tests (Figure 3A,B, respectively). In addition, the Egger’s test also shows no publication bias (*p* = 0.17) for the studies included in the high-level exposure group (Appendix A). Because only two studies were included in the low-level exposure group, Egger’s test could not be performed.

## 4. Discussion

The present study retrieved worldwide studies reporting the relationship between UCd concentrations and BMD or osteoporosis in postmenopausal women and conducted a meta-analysis of the original data to explore the association between environmental Cd exposure and risk of osteoporosis. To the best of our knowledge, this is the first global systematic study investigating whether environmental exposure to Cd is a risk factor for postmenopausal osteoporosis. Women in our study were at least 50 years old, which is the median age of natural menopause onset [45,46], and more than 90% of those included in the meta-analysis had confirmed postmenopausal status (Table 1). Women also had no known history of occupational Cd exposure, ensuring that the environment was the only source of Cd exposure. Furthermore, because tobacco use increases the body’s burden to Cd and is also a risk factor for osteoporosis [47], postmenopausal women were largely non-smokers (61 to 99%) to preclude any smoking-related impact of Cd exposure. In this regard, it is worth noting that smoking has been shown to have no effect on the relationship between UCd and BMD or osteoporosis [14,43,44].

Based on the inclusion criteria, ten studies with a total of 9205 women were included in this systematic review. Study quality was not considered to be a potential source of heterogeneity, as all studies were of high quality. The studies were conducted in different geographical regions, including East Asian countries such as China [23,39], Japan [25], South Korea [26], Western Europe, primarily Sweden [41,42,43], and North America [44]. According to the reported data, the studies reflected two distinct exposure scenarios. Postmenopausal women in Asian countries had approximately ten times higher UCd values than women in Europe or the United States, indicating that they were exposed to higher environmental levels of Cd. This is consistent with the fact that many areas in East Asian countries, such as China, are heavily polluted by Cd. Indeed, the Chinese agricultural soil and water resources are contaminated by Cd from industrial activities, mining, smelting, intensive use of Cd-containing fertilizers, and a high geological background of Cd [48,49], which is taken up by crops and vegetables and enters the human body through the food chain. Therefore, the observed disparity in UCd levels can be attributed to differences in dietary Cd exposure, as ingestion of contaminated food is the main source of Cd exposure in the non-smoking population. In line with this, food analysis revealed that Asian food products (primarily rice) contain higher levels of Cd than European or North American food [50,51].

Our systematic review supports the notion that higher UCd levels are associated with lower bone density in postmenopausal women, since only one study [26] failed to show an association (Table 1). Furthermore, the results of the meta-analysis showed that increased urinary levels of Cd were associated with increased risk of postmenopausal osteoporosis. Indeed, postmenopausal women with a low-level environmental exposure to Cd had a 95% increased risk of osteoporosis, whereas those with a high-level environmental exposure had a 99% increased risk, when compared with the respective reference groups with lower UCd levels. These results show that the risk of osteoporosis in the high-level exposure group was comparable to that in the low-level exposure group, though previous research using UCd stratified levels has found a dose-dependent increase in osteoporosis risk [23,41]. This discrepancy is likely related to the diverse populations (Caucasian versus Asian) included in our study that may have been influenced by genetic factors. In any case, a major finding in this study was that Cd exposure affects bone mass and increases the risk of osteoporosis even at low environmental levels.

The biological mechanisms by which Cd exerts its toxic effect on bone structure are complex and have not been fully elucidated yet. Osteoporosis caused by Cd may be associated with Cd-induced kidney damage by decreasing renal tubular reabsorption and increasing the urinary excretion of elements such as calcium and phosphorus, which are critical for maintaining bone metabolism and health [52]. Cd also decreases vitamin D [1,25(OH)2D] synthesis in kidneys, reducing the uptake of calcium in the gastrointestinal tract [53]. In addition, current evidence supports that Cd has a direct osteotoxic effect that can occur independently of renal dysfunction [54]. Bone tissue homeostasis is maintained by a balance between osteoblast-mediated bone formation and osteoclast-mediated bone resorption. An imbalance in bone remodeling leads to bone loss and osteoporosis [55]. Cd has been shown to directly disrupt the differentiation and metabolism of osteoblasts and their precursors, stimulate osteoclasts formation and activity, interfere with the production of bone collagen, and accelerate bone remodeling [13,52,56,57,58,59,60]. The mechanisms underlying the detrimental effects of Cd on bone metabolism are not completely understood, but they are likely to include cellular senescence, oxidative stress, DNA damage, mitochondrial dysfunction, apoptosis, and autophagy [61,62,63,64,65,66]. At the molecular level, Wu et al. [60] demonstrated that Cd suppresses osteogenic differentiation of bone marrow mesenchymal stem cells by inhibiting the canonical Wingless-related integration site (Wnt)/β-catenin pathway, which is known to play a crucial role in bone development and homeostasis, specially by modulating progenitor cells proliferation and differentiation [55]. Another master signaling pathway that regulates bone tissue metabolism is the phosphatidylinositol 3-kinase (PI3K)/Akt pathway [66]. Activation of the PI3K/Akt pathway has been shown to stimulate osteoblast proliferation and differentiation while also influencing osteoclast formation. Importantly, recent research by Ma et al. [56,65] has highlighted the role of the PI3K/Akt pathway in Cd-induced osteoporosis. The authors demonstrated in animal models and in vitro that Cd causes osteoporosis by suppressing PI3K/Akt-mediated osteoblast and osteoclast differentiation. Other signaling pathways, however, may be involved in Cd-induced osteoporosis, which requires further investigation. 

Women are at greater risk of developing Cd toxicity than men [26,40], especially after menopause. Because Cd absorption in the intestine is primarily mediated by the ferrous iron transporter [67], uptake of this metal is increased prior to menopause due to the low iron stores commonly observed in women of childbearing age. As women’s iron requirements decrease during menopause, so will their absorption of dietary Cd. However, because this coincides with the peak of renal Cd, health effects from exposure may occur at this time [68]. Estrogen depletion caused by menopause might be an important factor influencing the bone effects of Cd in women. Moreover, the presence of Cd may accelerate bone loss and cause osteoporosis together with other cofactors. Therefore, Cd exposure may be a major risk factor for this population group. In this regard, some studies were already conducted to provide a reference for risk assessment of Cd exposure in the female population. Suwazono et al. (2010) estimated benchmark dose of UCd for osteoporosis in a Swedish female population aged 53–64 years to be 1.8 and 2.9 µg/g creatinine, with their 95% lower confidence limits (BMDL) of 1.0 and 1.6 µg/g creatinine, for benchmark responses of 5% and 10%, respectively [43]. A similar study performed by Chen et al. (2013) based on a Chinese female population aged 40–86 years showed that the benchmark dose of UCd concentration related to osteoporosis was 5.30 and 9.06 µg/g creatinine and BMDL of 3.78 and 6.36 μg/g creatinine, respectively, for the same benchmark responses [69], which was much higher than those reported in the Swedish population. However, in a more recent study by Lv et al. (2017) [23], also conducted in a Chinese postmenopausal female population, the calculated benchmark dose of 0.64 and 1.77 µg/g creatinine and BMDL of 0.17 and 0.76 µg/g creatinine were much lower than those in both previous studies. Importantly, the benchmark dose for a 5% additional risk of osteoporosis was 0.64 µg/g creatinine which is consistent with the UCd cut-off value of 0.5 µg/g creatinine used in the present study for the low-level exposure group. Nonetheless, because of the significant differences in benchmark doses among the populations studied, more research in this area is mandatory.

Overall, the present study provides novel insights into the association between environmental Cd exposure and osteoporosis risk in postmenopausal women; however, it has significant limitations that must be recognized. First, there were a small number of studies that met our meta-analysis inclusion criteria, which could imply that the observed significant relationship between Cd exposure and osteoporosis is not sufficiently robust. Second, the cross-sectional design of the included studies precludes any inference of causality. However, it should be noted at this point that UCd is a biomarker of long-term Cd exposure, indicating accumulation in the body over years prior to BMD assessment. Third, the studies included in this systematic review may be affected by heterogeneity among methodologies and detection sites of BMD measurements, population genetic characteristics, among other factors. Finally, the heterogeneity of confounder adjustment strategies of the included studies may lead to additional bias for pooled effect size in meta-analyses. Therefore, our findings should be interpreted with caution, and further confirmation using large, high-quality prospective studies are required. In this regard, a prospective study published this year (after the end of our data collection period) in a cohort of Swedish postmenopausal women found that long-term exposure to very low levels of environmental Cd (median UCd concentrations of 0.33 µg/g creatinine) increased the risk of fractures, supporting our findings that environmental Cd exposure is a risk factor for bone damage even at low levels.

Finally, it must be noted that Cd accumulates gradually in the human body and, therefore, preventive strategies to reduce the environmental exposure to Cd and the associated burden of disease must be implemented early in life. In line with this, Chen [36] and Horiguchi [25] revealed that UCd levels remained high in residents of formerly polluted areas even after exposure was discontinued for more than ten years, implying that adverse effects continue even after exposure has stopped. Hence, Cd pollution control measures must be immediately strengthened if this risk factor is to be reduced for future generations.

## 5. Conclusions

The present study adds support to the evidence that environmental Cd exposure, even at low levels, is a risk factor for osteoporosis in postmenopausal women. However, further research with well-designed prospective studies is needed to validate this conclusion. Because this is a preventable risk factor, global environmental policy programs aimed specifically at reducing dietary Cd exposure have the potential to reduce the burden of postmenopausal osteoporosis in future generations. 

## Figures and Tables

**Figure 1 ijerph-20-00485-f001:**
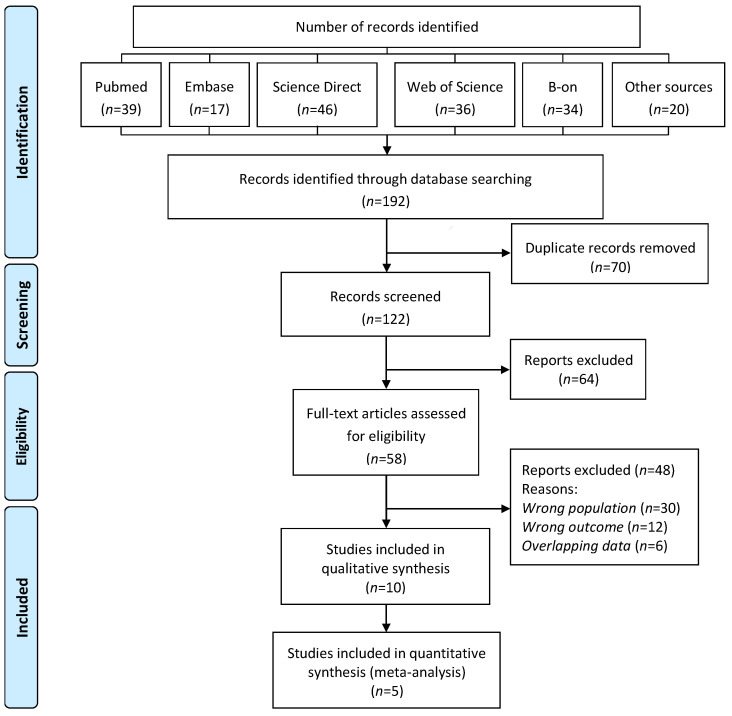
PRISMA flow diagram for the systematic review on the association between environmental exposure to cadmium and risk of osteoporosis in postmenopausal women.

**Figure 2 ijerph-20-00485-f002:**
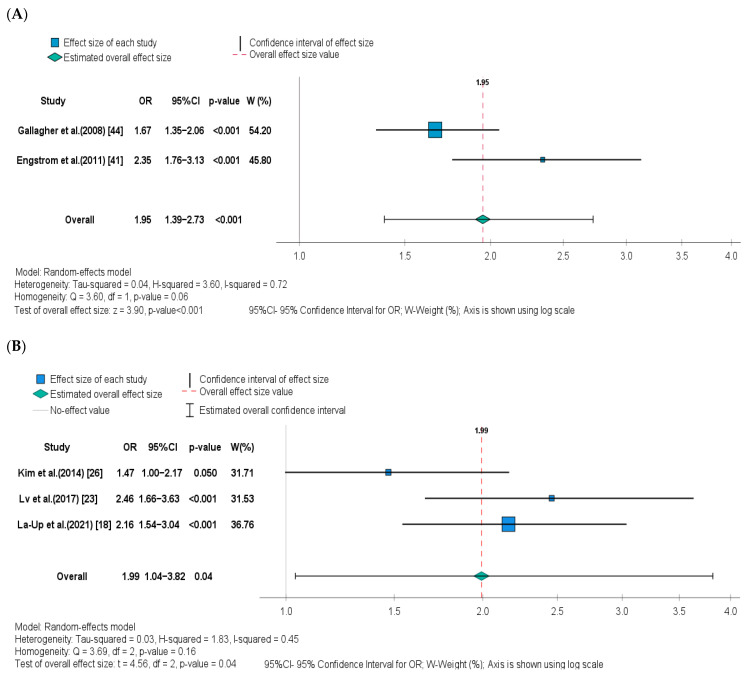
Forest plot and pooled effect estimates of the association between UCd levels and risk of osteoporosis in the (**A**) low-level Cd exposure group [41,44], and (**B**) high-level Cd exposure group [18,23,26].

**Figure 3 ijerph-20-00485-f003:**
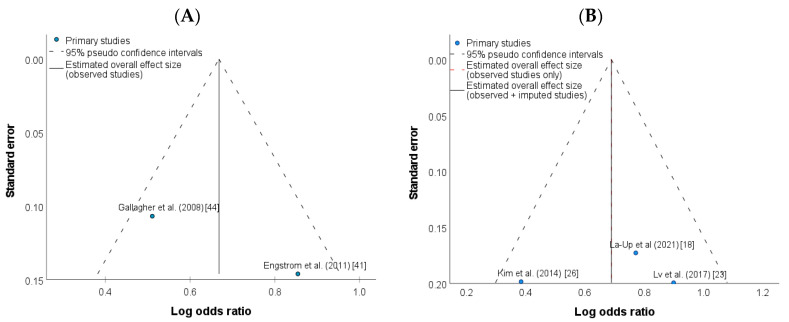
Funnel plot of studies investigating the association between UCd levels and risk of osteoporosis in the (**A**) low-level Cd exposure group [41,44], and (**B**) high-level Cd exposure group [18,23,26].

## Data Availability

Not applicable.

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
