# Peer review of "Association between Environmental Cadmium Exposure and Osteoporosis Risk in Postmenopausal Women: A Systematic Review and Meta-Analysis"

_ijerph, 2022, doi:10.3390/ijerph20010485_

Round 1
Reviewer 1 Report
The current meta-analysis study to find the association between environmental cadmium exposure and osteoporosis risk in postmenopausal women is worthy and detailed. As this disease is widespread worldwide, so the study will attract a lot of researchers to work on the issue. Here I have some minor concerns that are necessary to polish the manuscript.
1. Is environmental level cadmium can cause the same issues, e.g., fracture. If that is so, please highlight it in the manuscript.
2. The problem is clearly stated, and relevant literature is presented in support. But still, more information would make interesting reading. The interpretation of the results in the discussion part needs to be more in depth, with some latest references
3. Manuscript has some grammatical mistakes; author should revise it thoroughly.
4. Author didn’t compare results clearly in discussion section. Better to revise it accordingly as there are an adequate number of references in this section too.
5. From my point of view, author should include some data from such studies that used animal models for cd exposure, leading to osteoarthritis. This kind of data will be more helpful for clarity.
6. How environment-relevant cd enters body, author also need to give an overview in introduction section in a more brief way.
7. Researchers declared some elements as carcinogenic in animals but not in humans. Is the ratio of carcinogenicity the same in the cd?, if we look through one health aspect, this issue should be addressed and included in the current meta analysis.
8. How were the studies on different concentrations (or ranges of concentrations) selected to know how environmentally relevant are they?
Author Response
Reviewer #1
The current meta-analysis study to find the association between environmental cadmium exposure and osteoporosis risk in postmenopausal women is worthy and detailed. As this disease is widespread worldwide, so the study will attract a lot of researchers to work on the issue. Here I have some minor concerns that are necessary to polish the manuscript.
Author’s reply:
Firstly, we would like to thank Reviewer 1 for the comments and suggestions that helped to improve the quality of the manuscript. Below is an item-by-item reply to the Reviewer’s 1 comments and suggestions.
- Is environmental level cadmium can cause the same issues, e.g., fracture. If that is so, please highlight it in the manuscript.
Author’s reply:
We thank the reviewer for this comment, and we state in the revised manuscript that “population-based studies have shown that Cd exposure, even at low environmental levels, may result in decreased bone mass and an increased risk of osteoporosis and bone fractures [14, 23, 24]”.
- The problem is clearly stated, and relevant literature is presented in support. But still, more information would make interesting reading. The interpretation of the results in the discussion part needs to be more in depth, with some latest references.
Author’s reply:
We appreciate the reviewer's observation, which was also made by reviewer #2. In the revised manuscript, we improved the discussion section to address the mechanisms underlying cadmium-induced bone damage in greater depth, as follows:
Page 10, lines 309-330: “Bone tissue homeostasis is maintained by a balance between osteoblast-mediated bone formation and osteoclast-mediated bone resorption. An imbalance in bone remodeling leads to bone loss and osteoporosis [55]. Cd has been shown to directly disrupt the differentiation and metabolism of osteoblasts and their precursors, stimulate osteoclasts formation and activity, interfere with the production of bone collagen, and accelerate bone remodeling [13, 52, 56-60]. The mechanisms underlying the detrimental effects of Cd on bone metabolism are not completely understood, but they are likely to include cellular senescence, oxidative stress, DNA damage, mitochondrial dysfunction, apoptosis, and autophagy [61-66]. At the molecular level, Wu et al. [60] demonstrated that Cd suppresses osteogenic differentiation of bone marrow mesenchymal stem cells by inhibiting the canonical Wingless-related integration site (Wnt)/β-catenin pathway, which is known to play a crucial role in bone development and homeostasis, specially by modulating progenitor cells proliferation and differentiation [55]. Another master signaling pathway that regulates bone tissue metabolism is the phosphatidylinositol 3-kinase (PI3K)/Akt pathway [66]. Activation of the PI3K/Akt pathway has been shown to stimulate osteoblast proliferation and differentiation while also influencing osteo-clast formation. Importantly, recent research by Ma et al. [56, 65] has highlighted the role of the PI3K/Akt pathway in Cd-induced osteoporosis. The authors demonstrated in animal models and in vitro that Cd causes osteoporosis by suppressing PI3K/Akt-mediated osteoblast and osteoclast differentiation. Other signaling pathways, however, may be involved in Cd-induced osteoporosis, which requires further investigation.”
References have also been updated as requested.
New added references:
- Amjadi-Moheb, F.; Akhavan-Niaki, H., Wnt signaling pathway in osteoporosis: Epigenetic regulation, interaction with other signaling pathways, and therapeutic promises. Journal of Cellular Physiology 2019, 234 (9), 14641-14650.
- Chen, X.; Ren, S.; Zhu, G.; Wang, Z.; Wen, X., Emodin suppresses cadmium-induced osteoporosis by inhibiting osteoclast formation. Environ Toxicol Pharmacol 2017, 54, 162-168.
- Wu, L.; Wei, Q.; Lv, Y.; Xue, J.; Zhang, B.; Sun, Q.; Xiao, T.; Huang, R.; Wang, P.; Dai, X.; Xia, H.; Li, J.; Yang, X.; Liu, Q., Wnt/β-Catenin Pathway Is Involved in Cadmium-Induced Inhibition of Osteoblast Differentiation of Bone Marrow Mesenchymal Stem Cells. Int J Mol Sci 2019, 20 (6).
- Ou, L.; Wang, H.; Wu, Z.; Wang, P.; Yang, L.; Li, X.; Sun, K.; Zhu, X.; Zhang, R., Effects of cadmium on osteoblast cell line: Exportin 1 accumulation, p-JNK activation, DNA damage and cell apoptosis. Ecotoxicol Environ Saf 2021, 208, 111668.
- Liu, W.; Dai, N.; Wang, Y.; Xu, C.; Zhao, H.; Xia, P.; Gu, J.; Liu, X.; Bian, J.; Yuan, Y.; Zhu, J.; Liu, Z., Role of autophagy in cadmium-induced apoptosis of primary rat osteoblasts. Sci Rep 2016, 6, 20404.
- Luo, H.; Gu, R.; Ouyang, H.; Wang, L.; Shi, S.; Ji, Y.; Bao, B.; Liao, G.; Xu, B., Cadmium exposure induces osteoporosis through cellular senescence, associated with activation of NF-κB pathway and mitochondrial dysfunction. Environ Pollut 2021, 290, 118043.
- Ran, D.; Ma, Y.; Liu, W.; Luo, T.; Zheng, J.; Wang, D.; Song, R.; Zhao, H.; Zou, H.; Gu, J.; Yuan, Y.; Bian, J.; Liu, Z., TGF-β-activated kinase 1 (TAK1) mediates cadmium-induced autophagy in osteoblasts via the AMPK / mTORC1 / ULK1 pathway. Toxicology 2020, 442, 152538.
- Ma, Y.; Ran, D.; Cao, Y.; Zhao, H.; Song, R.; Zou, H.; Gu, J.; Yuan, Y.; Bian, J.; Zhu, J.; Liu, Z., The effect of P2X7 on cadmium-induced osteoporosis in mice. J Hazard Mater 2021, 405, 124251.
- Zheng, J.; Zhuo, L.; Ran, D.; Ma, Y.; Luo, T.; Zhao, H.; Song, R.; Zou, H.; Zhu, J.; Gu, J.; Bian, J.; Yuan, Y.; Liu, Z., Cadmium induces apoptosis via generating reactive oxygen species to activate mitochondrial p53 pathway in primary rat osteoblasts. Toxicology 2020, 446, 152611.
- Fujishiro, H.; Hamao, S.; Tanaka, R.; Kambe, T.; Himeno, S., Concentration-dependent roles of DMT1 and ZIP14 in cadmium absorption in Caco-2 cells. J Toxicol Sci 2017, 42 (5), 559-567.
- Manuscript has some grammatical mistakes; author should revise it thoroughly.
Author’s reply:
As suggested by the reviewer, we have gone through the revised article and corrected minor typing mistakes and grammatical errors.
- Author didn’t compare results clearly in discussion section. Better to revise it accordingly as there are an adequate number of references in this section too.
Author’s reply: We have only highlighted the major findings from our study in the discussion to avoid repetitions between the results and discussion sections. Nonetheless, as previously stated, the discussion in the revised paper has been improved.
- From my point of view, author should include some data from such studies that used animal models for cd exposure, leading to osteoarthritis. This kind of data will be more helpful for clarity.
Author’s reply:
We thank the Reviewer for this input that has now been considered in the revised manuscript.
- How environment-relevant cd enters body, author also need to give an overview in introduction section in a more brief way.
Author’s reply:
Cadmium in the environment can enter the human body mainly via the food chain (diet) or cigarette smoking. We stated in the manuscript that “the general population is primarily exposed to Cd through contaminated food (mostly cereals and leafy vegetables) and water [18]. Additionally, Cd can enter the human body via other sources, including tobacco smoking (because tobacco plants absorb Cd from the soil and concentrate it in the leaves) and occupational exposure to Cd fumes or dust in the workplace [19].”
We also included a brief mention of the main transporter mediating Cd intestinal absorption, as follows:
“Because Cd absorption in the intestine is primarily mediated by the ferrous iron transporter [67], uptake of this metal is increased prior to menopause due to the low iron stores commonly observed in women of childbearing age.“
New added reference:
- Fujishiro, H.; Hamao, S.; Tanaka, R.; Kambe, T.; Himeno, S., Concentration-dependent roles of DMT1 and ZIP14 in cad-mium absorption in Caco-2 cells. J Toxicol Sci 2017, 42 (5), 559-567.
- Researchers declared some elements as carcinogenic in animals but not in humans. Is the ratio of carcinogenicity the same in the cd?, if we look through one health aspect, this issue should be addressed and included in the current meta analysis.
Author’s reply:
Cadmium is classified as a Group I human carcinogen by the International Agency for Research on Cancer (IARC). Furthermore, epidemiologic evidence suggests that occupational or environmental Cd exposure is associated with the development of lung, breast, prostate, pancreas, urinary bladder, and nasopharynx cancers (Mezynska and Brzóska, 2018).
We agree with the reviewer that a systematic review on the carcinogenic effect of Cd is important because of the public health implications, but there is insufficient data from the included studies to present it here. It's a great idea for future large-scale systematic reviews and meta-analyses.
Reference:
Mezynska, M.; Brzóska, M. M., Environmental exposure to cadmium-a risk for health of the general population in industrialized countries and preventive strategies. Environ Sci Pollut Res Int 2018, 25 (4), 3211-3232.
- How were the studies on different concentrations (or ranges of concentrations) selected to know how environmentally relevant are they?
Author’s reply:
All the studies included in this systematic review were conducted on female populations exposed to realistic Cd concentrations in the environment. Furthermore, because women had no known history of occupational Cd exposure, the environment was the only source of Cd exposure. This statement was included in the manuscript.

Reviewer 2 Report
The paper regarding Association between environmental cadmium exposure and osteoporosis risk in postmenopausal women: a systematic review and meta-analysis is well presented.
The authors showed . The study is well presented.
There are some possible issues
Previous studies have found that bone remodelling is regulated by an intimate cross-talk among osteoclasts and osteoblasts as well as other cell types including osteocytes, bone lining cells, osteomacs, and vascular endothelial cells. It is not sure if cadmium might affect these cell types due to environmental exposure? It would be relevant to discuss this possibility by referring to this study.
It would be relevant to present how cadmium might affect bone cells in a figure or table?
At the molecular level, cadmium is found to be involved in the regulation of Jun N terminal kinase (JNK)/nuclear factor kappa B (NF-κB) and phosphatidylinositol 3-kinases (PI3K)/protein kinase (AKT) pathways (for example PMID: 36375572), which are also important for osteoclast signallings and osteoclast related osteoporosis (for example, PMID: 22087256, PMID: 35999378, PMID: 35988868, PMID: 33782965). It would be informative to discuss the role of cadmium in osteoclast related signalling, which would enhance the overview and potential of this study.
There are some possible typos
astudy included in meta-analysis?? A study ??
Odds ratio (OR) with 95% confidence interval (CI) were estimated?? Were should be Was ??
Author Response
Reviewer #2
The paper regarding Association between environmental cadmium exposure and osteoporosis risk in postmenopausal women: a systematic review and meta-analysis is well presented.
The authors showed . The study is well presented.
Author’s reply:
Firstly, we would like to thank Reviewer 2 for the comments and suggestions that helped to improve the quality of the manuscript. Below is an item-by-item reply to the Reviewer’s 2 comments and suggestions.
There are some possible issues
Previous studies have found that bone remodelling is regulated by an intimate cross-talk among osteoclasts and osteoblasts as well as other cell types including osteocytes, bone lining cells, osteomacs, and vascular endothelial cells. It is not sure if cadmium might affect these cell types due to environmental exposure? It would be relevant to discuss this possibility by referring to this study.
Author’s reply:
We thank the reviewer for this insightful comment, and we have now discussed the impact of Cd in bone remodeling at the cellular level, as follows:
Page 5, line 213: “Bone tissue homeostasis is maintained by a balance between osteoblast-mediated bone formation and osteoclast-mediated bone resorption. An imbalance in bone remodeling leads to bone loss and osteoporosis [55]. Cd has been shown to directly disrupt the differentiation and metabolism of osteoblasts and their precursors, stimulate osteoclasts formation and activity, interfere with the production of bone collagen, and accelerate bone remodeling [13, 52, 56-60]. The mechanisms underlying the detrimental effects of Cd on bone metabolism are not completely understood, but they are likely to include cellular senescence, oxidative stress, DNA damage, mitochondrial dysfunction, apoptosis, and autophagy [61-66].“
New added references:
- Amjadi-Moheb, F.; Akhavan-Niaki, H., Wnt signaling pathway in osteoporosis: Epigenetic regulation, interaction with other signaling pathways, and therapeutic promises. Journal of Cellular Physiology 2019, 234 (9), 14641-14650.
- Chen, X.; Ren, S.; Zhu, G.; Wang, Z.; Wen, X., Emodin suppresses cadmium-induced osteoporosis by inhibiting osteoclast formation. Environ Toxicol Pharmacol 2017, 54, 162-168.
- Wu, L.; Wei, Q.; Lv, Y.; Xue, J.; Zhang, B.; Sun, Q.; Xiao, T.; Huang, R.; Wang, P.; Dai, X.; Xia, H.; Li, J.; Yang, X.; Liu, Q., Wnt/β-Catenin Pathway Is Involved in Cadmium-Induced Inhibition of Osteoblast Differentiation of Bone Marrow Mesenchymal Stem Cells. Int J Mol Sci 2019, 20 (6).
- Ou, L.; Wang, H.; Wu, Z.; Wang, P.; Yang, L.; Li, X.; Sun, K.; Zhu, X.; Zhang, R., Effects of cadmium on osteoblast cell line: Exportin 1 accumulation, p-JNK activation, DNA damage and cell apoptosis. Ecotoxicol Environ Saf 2021, 208, 111668.
- Liu, W.; Dai, N.; Wang, Y.; Xu, C.; Zhao, H.; Xia, P.; Gu, J.; Liu, X.; Bian, J.; Yuan, Y.; Zhu, J.; Liu, Z., Role of autophagy in cadmium-induced apoptosis of primary rat osteoblasts. Sci Rep 2016, 6, 20404.
- Luo, H.; Gu, R.; Ouyang, H.; Wang, L.; Shi, S.; Ji, Y.; Bao, B.; Liao, G.; Xu, B., Cadmium exposure induces osteoporosis through cellular senescence, associated with activation of NF-κB pathway and mitochondrial dysfunction. Environ Pollut 2021, 290, 118043.
- Ran, D.; Ma, Y.; Liu, W.; Luo, T.; Zheng, J.; Wang, D.; Song, R.; Zhao, H.; Zou, H.; Gu, J.; Yuan, Y.; Bian, J.; Liu, Z., TGF-β-activated kinase 1 (TAK1) mediates cadmium-induced autophagy in osteoblasts via the AMPK / mTORC1 / ULK1 pathway. Toxicology 2020, 442, 152538.
- Ma, Y.; Ran, D.; Cao, Y.; Zhao, H.; Song, R.; Zou, H.; Gu, J.; Yuan, Y.; Bian, J.; Zhu, J.; Liu, Z., The effect of P2X7 on cadmium-induced osteoporosis in mice. J Hazard Mater 2021, 405, 124251.
- Zheng, J.; Zhuo, L.; Ran, D.; Ma, Y.; Luo, T.; Zhao, H.; Song, R.; Zou, H.; Zhu, J.; Gu, J.; Bian, J.; Yuan, Y.; Liu, Z., Cadmium induces apoptosis via generating reactive oxygen species to activate mitochondrial p53 pathway in primary rat osteoblasts. Toxicology 2020, 446, 152611.
It would be relevant to present how cadmium might affect bone cells in a figure or table?
Author’s reply:
We understand the reviewer's point of view, but the authors believe that a table containing a detailed examination of the mechanisms underlying Cd-induced bone damage is not the focus of the current systematic review and should be addressed in a separate study in the future. We will, however, provide it if the reviewer continues to request it.
At the molecular level, cadmium is found to be involved in the regulation of Jun N terminal kinase (JNK)/nuclear factor kappa B (NF-κB) and phosphatidylinositol 3-kinases (PI3K)/protein kinase (AKT) pathways (for example PMID: 36375572), which are also important for osteoclast signallings and osteoclast related osteoporosis (for example, PMID: 22087256 Versatile roles of V-ATPases accessory subunit Ac45 in osteoclast formation and function, PMID: 35999378, PMID: 35988868, PMID: 33782965). It would be informative to discuss the role of cadmium in osteoclast related signalling, which would enhance the overview and potential of this study.
Author’s reply:
We appreciate the Reviewer's input, and we have now addressed the putative molecular mechanisms involved in Cd-induced osteoporosis, as follows:
Page 10, lines 317-330: “At the molecular level, Wu et al. [60] demonstrated that Cd suppresses osteogenic differentiation of bone marrow mesenchymal stem cells by inhibiting the canonical Wingless-related integration site (Wnt)/β-catenin pathway, which is known to play a crucial role in bone development and homeostasis, specially by modulating progenitor cells proliferation and differentiation [55]. Another master signaling pathway that regulates bone tissue metabolism is the phosphatidylinositol 3-kinase (PI3K)/Akt pathway [66]. Activation of the PI3K/Akt pathway has been shown to stimulate osteoblast proliferation and differentiation while also influencing osteo-clast formation. Importantly, recent research by Ma et al. [56, 65] has highlighted the role of the PI3K/Akt pathway in Cd-induced osteoporosis. The authors demonstrated in animal models and in vitro that Cd causes osteoporosis by suppressing PI3K/Akt-mediated osteoblast and osteoclast differentiation. Other signaling pathways, however, may be involved in Cd-induced osteoporosis, which requires further investigation.”
New added references:
- Amjadi-Moheb, F.; Akhavan-Niaki, H., Wnt signaling pathway in osteoporosis: Epigenetic regulation, interaction with other signaling pathways, and therapeutic promises. Journal of Cellular Physiology 2019, 234 (9), 14641-14650.
- Wu, L.; Wei, Q.; Lv, Y.; Xue, J.; Zhang, B.; Sun, Q.; Xiao, T.; Huang, R.; Wang, P.; Dai, X.; Xia, H.; Li, J.; Yang, X.; Liu, Q., Wnt/β-Catenin Pathway Is Involved in Cadmium-Induced Inhibition of Osteoblast Differentiation of Bone Marrow Mesenchymal Stem Cells. Int J Mol Sci 2019, 20 (6).
- Ran, D.; Ma, Y.; Liu, W.; Luo, T.; Zheng, J.; Wang, D.; Song, R.; Zhao, H.; Zou, H.; Gu, J.; Yuan, Y.; Bian, J.; Liu, Z., TGF-β-activated kinase 1 (TAK1) mediates cadmium-induced autophagy in osteoblasts via the AMPK / mTORC1 / ULK1 pathway. Toxicology 2020, 442, 152538.
- Ma, Y.; Ran, D.; Cao, Y.; Zhao, H.; Song, R.; Zou, H.; Gu, J.; Yuan, Y.; Bian, J.; Zhu, J.; Liu, Z., The effect of P2X7 on cadmium-induced osteoporosis in mice. J Hazard Mater 2021, 405, 124251.
- Zheng, J.; Zhuo, L.; Ran, D.; Ma, Y.; Luo, T.; Zhao, H.; Song, R.; Zou, H.; Zhu, J.; Gu, J.; Bian, J.; Yuan, Y.; Liu, Z., Cadmium induces apoptosis via generating reactive oxygen species to activate mitochondrial p53 pathway in primary rat osteoblasts. Toxicology 2020, 446, 152611.
“There are some possible typos
astudy included in meta-analysis?? A study ??
Odds ratio (OR) with 95% confidence interval (CI) were estimated?? Were should be Was ??”
Author’s reply:
Corrections were done. Thank you.
